# Running and Jumping After Muscle Fatigue in Subjects with a History of Knee Injury: What Are the Acute Effects of Wearing a Knee Brace on Biomechanics?

**DOI:** 10.3390/bioengineering12060661

**Published:** 2025-06-16

**Authors:** Tobias Heß, Thomas L. Milani, Jan Stoll, Christian Mitschke

**Affiliations:** 1Department of Movement Science for Prevention and Rehabilitation, Institute of Human Movement Science and Health, Chemnitz University of Technology, Thüringer Weg 11, 09126 Chemnitz, Germanychristian.mitschke@hsw.tu-chemnitz.de (C.M.); 2Research Group for Biomechanics and Sensory Function, Institute of Human Movement Science and Health, Chemnitz University of Technology, Thüringer Weg 11, 09126 Chemnitz, Germany

**Keywords:** knee joint injury, muscle fatigue, knee brace, running, counter-movement jump, knee joint moment, knee joint stability, pain, rehabilitation

## Abstract

The knee is one of the most frequently injured joints, involving various structures. To prevent reinjury after rehabilitation, braces are commonly used. However, most studies on knee supports focus on subjects with anterior cruciate ligament (ACL) injuries and do not account for muscle fatigue, which typically occurs during prolonged intense training and can significantly increase the risk of injury. Hence, this study investigates the acute effects of wearing a knee brace on biomechanics in subjects with a history of various unilateral knee injuries or pain under muscle fatigue. In total, 50 subjects completed an intense fatigue protocol and then performed counter-movement jumps and running tests on a force plate while tracking kinematics with a marker-based 3D motion analysis system. Additionally, subjects filled out a visual analog scale (VAS) to assess knee pain and stability. Tests were conducted on the injured leg with and without a knee brace (Sports Knee Support, Bauerfeind AG, Zeulenroda-Triebes, Germany) and on the healthy leg. Results indicated that wearing the knee brace stabilized knee movement in the frontal plane, with a significant reduction in maximal medio-lateral knee acceleration and knee abduction moment during running and jumping. The brace also normalized loading on the injured leg. We observed higher maximal knee flexion moments, which were associated with increased vertical ground reaction forces, segment velocities, and knee flexion angles. Subjects reported less pain and greater stability while wearing the knee brace. Therefore, we confirm that wearing a knee brace on the injured leg improves joint biomechanics by enhancing stability and kinematics and reducing pain during running and jumping, even with muscle fatigue. Consequently, wearing a knee brace after a knee joint injury may reduce the risk of reinjury.

## 1. Introduction

The knee is one of the most frequently affected joints in the human body, with injuries involving various structures, including ligaments, menisci, cartilage, and bones [1,2,3,4]. In fact, knee injuries account for up to approx. 40% of all joint injuries in sports such as soccer, tennis, basketball, volleyball, and running [5,6,7,8,9]. Once the knee has been injured, the risk of reinjury increases significantly, with up to 30% of individuals sustaining a second injury, depending on the conditions [1,3,10,11,12]. Reinjuries commonly occur during prolonged or intense physical activities, as fatigue compromises muscle performance and neuromuscular control [2,11,13,14]. As a result, joint stability and function are compromised, increasing the risk of injury even during common movements such as running or jumping [15,16,17,18,19].

To reduce the risk of reinjuries and further damage to the knee joint, knee supports such as braces are frequently used. They provide external mechanical support through the elastic material and integrated lateral and medial rubber bands. Therefore, they promote better joint alignment, enhance joint stability and muscular function, reduce pain, and might improve proprioception through compression of the underlying musculoskeletal structures [15,16,20,21]. However, most studies investigating the effects of knee supports on running and jumping have either focused on healthy subjects [14,15,17,18,19,20,22,23,24,25,26,27,28] or predominantly on subjects with anterior cruciate ligament (ACL) injuries [16,29,30,31,32,33,34,35,36]. Therefore, the effect of knee supports on the biomechanics of subjects with injured knee structures other than the ACL remains largely unclear. Moreover, none of these studies investigated subjects under conditions of muscle fatigue, which typically occurs during prolonged intense physical activities [2,11,13]. Most studies also used treadmills for walking or running [15,16,19,22,32,37,38], instead of overground walking or running on a force plate [20,21,30]. Although using a treadmill for biomechanical investigations mostly offers methodological advantages, it does not represent the natural running movement and shows various differences in kinetic and kinematic parameters compared to overground running [39,40,41,42].

Hence, in this study, we aimed to investigate the acute effects of wearing a knee brace on knee joint biomechanics in subjects with a history of various unilateral knee injuries or pain under conditions of muscle fatigue. We hypothesize that wearing a knee brace on the injured leg improves joint biomechanics by enhancing stability and kinematics (e.g., knee flexion angle), normalizing knee joint loading (e.g., knee flexion and abduction moments), and reducing pain during running and jumping.

## 2. Materials and Methods

### 2.1. Subjects

In summary, 50 subjects with various unilateral knee joint injuries were recruited for this study (Table 1). The injuries must have occurred between 1 and 10 years prior to the examination. Subjects should experience discomfort, such as mild to moderate pain or the feeling of instability or giving way in the knee joint during activities like running or jumping. Subjects should be between 18 and 50 years old, should have received a verbal or written return-to-sport recommendation from a medical doctor, and, therefore, should still be actively engaged in sports for at least 2 h per week. Exclusion criteria were acute injuries of the knee joint or any other joint of the leg within the last 3 months, use of prostheses or endoprostheses, inflammatory joint diseases, neurological diseases or dysfunctions, cardiovascular diseases, and any other conditions affecting motor performance. Before the examination, all subjects were informed about the study's purpose and provided written informed consent. All procedures were conducted in accordance with the Declaration of Helsinki and received approval from the Ethics Committee of Chemnitz University of Technology (reference number #101508546).

### 2.2. Experimental Setup and Data Acquisition

All anthropometric data were collected, and 16 reflective markers (Plug-in Gait lower body marker set) were attached to the subjects' pelvis and both legs and feet for motion capturing (Figure 1). Subsequently, subjects performed a standardized fatigue protocol to induce muscle fatigue in the lower extremities. The fatigue protocol consisted of three consecutive sets, each performed at an intensity level adjusted to each subject’s individual fitness. Each set included 60 s of jumping jacks, 20–30 repetitions of calf raises, 10–20 repetitions of step-ups and side lunges, and 15–25 squat jumps. After completing the three sets, subjects performed a wall sit until physical exertion. Before and after performing the fatigue protocol, subjects rated their level of exertion using a Borg scale, which ranged from 6 (no exertion) to 20 (high exertion) [43]. Immediately after the fatigue protocol, subjects performed overground running and counter-movement jumps in randomized order on a force plate (Kistler, Winterthur, Switzerland; dimensions 0.6 × 0.9 m; sampling frequency 1000 Hz), while their kinematics were tracked using a 3D motion analysis system with 10 cameras (Vicon, Oxford, UK; sampling frequency 200 Hz, Nexus 2.10.2). For both running and jumping tests, five consecutive trials were conducted in a randomized order for each of the three measurement conditions: healthy leg, injured leg, and injured leg with a knee brace. After completing both the running and jumping tests, subjects filled out a visual analog scale (VAS) to assess knee pain and stability for each of the three measurement conditions, with ratings ranging from 0 (no pain, very stable) to 10 (extreme pain, very unstable), respectively (Table 2). All tests were conducted using the subjects' regular sports shoes. The knee brace used in this study was the “Sports Knee Support” brace (Bauerfeind AG, Zeulenroda-Triebes, Germany). According to the manufacturer, the brace is made of ultralight elastic knit material, which provides an alternating pressure massage during movement. Integrated lateral and medial rubber bands offer stability, while the embedded knitted elastic silicone pad securely positions and guides the kneecap, ensuring optimal force distribution within the knee joint.

#### Running and Jumping Tests

Before the running test, the individual running speed was determined for each subject. For this purpose, each subject ran five times over the force plate, starting approximately 7 m before it, while the running speed was measured using light barriers (ALGE-TIMING, Lustenau, Austria) (Figure 1A). From these five runs, the average speed was calculated, with ±5% as the upper and lower speed limits. If a subject exceeded their individual running speed limits or if the force plate was missed with the leg to be tested, the trials were repeated until five valid trials were collected for each measurement condition. 

For the counter-movement jumps, subjects were instructed to stand in front of the force plate with a hip-width stance, keeping their arms crossed on the chest to prevent the reflective markers on the hips from being obscured during measurement. Upon an acoustic signal, the subjects stepped onto the force plate with the leg to be tested. Immediately afterward, they performed the counter-movement jump by flexing the knee joint to approximately 90°, followed by an explosive push-off, fully straightening the legs and jumping as high as possible while keeping their arms still. After executing the jump, the subjects were required to stand upright again, wait for a few seconds, and then step off the force plate (Figure 1B). Trials were considered invalid and had to be repeated if the legs were bent during the jump, the jump went forward instead of upward, or if the foot of the tested leg did not land on the force plate completely. For each measurement condition, five trials were collected.

### 2.3. Statistical Analysis

Biomechanical parameters were extracted from force plate and Vicon data using a custom routine written in MATLAB R2023b (MathWorks™, Natick, MA, USA). Depending on the task, this included the maximal vertical ground reaction force, ground contact time, maximal knee flexion angle, maximal knee flexion moment, maximal knee abduction moment, maximal medio-lateral knee acceleration, and maximal jump height. All parameters were calculated for the ground contact phase. For statistical analysis, the means and standard deviations (mean ± SD) were calculated for each biomechanical parameter based on the five trials for each subject. The Shapiro–Wilk test was used to assess normal distribution. To investigate subjects' fatigue induced by the fatigue protocol, as well as subjects' stability and pain, a sign test was used. Jumping heights before and after completing the protocol were compared using a *t*-test. To evaluate differences between measurement conditions (healthy leg, injured leg, injured leg with knee brace), a one-way ANOVA for repeated measures followed by Bonferroni post hoc tests was conducted for normally distributed data, while the Friedman test was used for non-normally distributed data. Statistical significance for all tests was set at α = 0.05. Effect sizes were calculated using Cohen's d and categorized as trivial (<0.2), small (<0.5), medium (<0.8), or large (≥0.8). Subjective data from the ratings were analyzed using the t-test for independent samples if normally distributed and the Mann–Whitney test if not normally distributed. 

## 3. Results

### 3.1. Demographic, Clinical, and Subjective Data

Table 1 presents the demographic and clinical characteristics of the subjects. The cohort included 50 participants, with a predominance of males (*n* = 31) over females (*n* = 19). The most frequently injured structures were ligaments and menisci, followed by cartilage damage, inflammations, and, lastly, fractures and bone injuries. These injuries occurred more frequently in the left leg (*n* = 28) than in the right leg (*n* = 22), and the mean time since injury was 6.1 ± 3.7 years. Most injuries were treated conservatively (*n* = 27) rather than surgically (*n* = 23).

Table 2 shows the effect of the fatigue protocol. After completing the protocol, subjects exhibited a significantly higher subjective level of physical exertion compared to before, which was also reflected by a significant reduction in jumping height.

The subjective ratings on the VAS indicated a significant reduction in pain and greater stability when wearing the knee brace on the injured leg while running. Subjects also reported significantly increased knee stability during jumping (Table 3).

### 3.2. Motor Performance

#### 3.2.1. Running

The average running speed across all subjects and measurement conditions was 3.3 ± 0.4 m/s. Several statistically significant differences were found between the measurement conditions for the running test. When comparing the healthy leg with the injured leg, the injured leg primarily showed lower maximal knee abduction moments, whereas the other parameters did not show statistically significant differences (Figure 2 and Table 4). The effect of the brace resulted in significantly increased maximal knee flexion moments, decreased maximal knee abduction moments, and significantly reduced maximal medio-lateral knee accelerations when comparing the injured leg with and without the knee brace (Figure 2). Moreover, wearing the brace on the injured leg resulted in significantly less ground contact time during running compared to the injured leg without the brace (Table 4).

#### 3.2.2. Counter-Movement Jump

For the jumping test, only the maximal knee flexion angle revealed statistically significant differences, with lower values for the injured leg compared to the healthy leg (Table 4). When comparing the injured leg without and with the brace, the effect of the brace was associated with significantly higher maximal knee flexion moments, reduced maximal medio-lateral knee acceleration (Figure 3), as well as higher maximal knee flexion angles (Table 5).

## 4. Discussion

In this study, we investigated the acute effects of wearing a knee brace on knee joint biomechanics in subjects with a history of various unilateral knee injuries or pain under conditions of muscle fatigue. We hypothesize that wearing a knee brace on the injured leg improves joint biomechanics by enhancing stability and kinematics (e.g., knee flexion angle), normalizing knee joint loading (e.g., knee flexion and abduction moments), and reducing pain during running and jumping.

### 4.1. Effects of the Injuries

The effect of the injury resulted in several differences between the healthy and injured legs during running and jumping. These included significantly lower maximal knee abduction moments and reduced vertical ground reaction forces. Since our subjects reported slight pain during the tests on the VAS scale (2.1 ± 1.4), this reduction in the maximal load may have been due to increased awareness and caution [16,21,29,30,31]. This could also explain why the injured leg exhibited shorter ground contact times during the stance phase of running. The injured leg also showed significantly reduced flexion during the ground contact phase, particularly during the landing phase of jumping. Besides pain, this may have been caused by the sensation of instability, as reported by the subjects on the VAS scale (2.0 ± 1.4). This seems plausible, as the collateral ligaments of the knee joint relax during knee flexion. Therefore, reduced flexion may help to minimize additional joint instability in the frontal plane [44,45,46]. Furthermore, injury-related muscle weakness might force subjects to reduce their flexion angle, as greater knee flexion generates higher moments that the muscles must counteract [47,48,49]. Besides impairments of the musculoskeletal system, the differences observed in kinetic and kinematic parameters between the injured and healthy legs might also be attributed to damage affecting various components of the proprioceptive system. This includes damage to specialized receptors located in the muscles, tendons, ligaments, and knee joint capsule, which provide crucial information about muscle length, tension, contraction speed, and joint position [50,51,52,53,54,55].

### 4.2. Effects of the Knee Brace

Wearing the brace altered several kinetic and kinematic parameters during running and jumping. The most significant change was the reduction in the maximal medio-lateral knee acceleration, which translates to improved stabilization of the knee joint in the frontal plane. This may primarily result from the mechanical support provided by the elastic knit material and the integrated lateral and medial rubber bands of the brace, which appear to play a crucial role in reducing relative motion between the femur and tibia [20,50]. However, improved joint stability might also be attributed to enhanced proprioception due to the compression of underlying musculoskeletal structures [26,30,50], as well as to altered muscle activity and improved muscular coordination of the knee joint [16,20,32,37,56]. Enhanced knee joint stability from wearing the brace was also reported by our subjects on the VAS scale. Similar results were also found in a recent study by Kitagawa et al., which investigated the effects of a soft knee brace on physical load and joint stability in healthy subjects during treadmill running using an IMU sensor. Their results showed that the soft knee brace significantly reduced the magnitude of medio-lateral shank acceleration [15]. Additionally, the study by Focke et al. found that in ACL-deficient subjects, soft knee braces reduced the maximum valgus angle compared to the unbraced condition, stabilizing the knee joint in the frontal plane during the stance phase of walking [30]. Another parameter in our study that showed a significant reduction due to wearing the brace in the frontal plane is the maximal knee abduction moment. This effect was especially pronounced during running when frontal plane knee joint moments are generally higher due to single-leg ground contact and the position of the center of mass relative to the knee joint, creating an abduction moment. Similar results have also been found in other studies when wearing a brace. For instance, Mougin et al. and Robert-Lachaine et al. reported reduced knee joint moments in the frontal plane at different walking speeds [16,21]. In the study by Moon et al., they reported that wearing a brace during drop jump tasks reduced the knee joint’s maximal abduction angles and adduction moments [17]. Therefore, controlling medio-lateral accelerations and frontal plane joint moments through the use of a brace may help to reduce the risk of ligament ruptures and secondary complications, such as osteoarthritis of the knee joint [2,16,21,30].

Regarding the sagittal plane, our results revealed significantly higher maximal knee flexion moments during both running and jumping when subjects wore the brace. Increased joint moments generally correspond to greater joint loading, which might be a result of the mechanical support generated by the brace during knee flexion [16,18,26,30]. This effect becomes clearer when examining the individual variables that define the joint moment. These include various kinematic and kinetic parameters, such as joint angle-dependent lever arms, ground reaction forces, and accelerations [57,58,59]. Hence, the increased flexion moments observed in our study might result from greater maximal knee flexion angles, slightly higher vertical ground reaction forces, and increased segment accelerations due to the shorter ground contact times, especially during running. Therefore, this suggests that subjects might voluntarily or involuntarily apply more weight to the ground and increase joint loading when wearing the brace. This is further supported by the fact that our subjects reported slightly less pain, particularly during running (2.1 ± 1.4 vs. 1.9 ± 1.3), and felt slightly more stable, especially during jumping (2.0 ± 1.4 vs. 1.9 ± 1.3) when wearing the brace. Nevertheless, the brace-induced changes in biomechanical parameters are more pronounced than those in pain and stability, suggesting that biomechanical measures may be more sensitive than subjective ratings.

Higher ground reaction forces and altered knee flexion angles due to bracing have also been found in various other studies. For instance, Butler et al. investigated the effects of knee bracing on knee joint function during jumping in subjects six months after ACL reconstruction. They found that on the braced surgical side, subjects exhibited increased knee flexion angles, higher flexion velocities, and greater ground reaction forces [33]. Baltaci et al. also reported that wearing knee braces led to greater force production during functional squat testing [26]. Several other studies have reported altered knee joint flexion angles when wearing knee braces, with changes ranging from approximately 1° to 5° across various activities, such as walking [16,19], running [19], and jumping [17,18,28]. Nevertheless, wearing the brace did not help our subjects jump higher, a finding that was confirmed by another study [18].

Due to differences in movement tasks, measured parameters, subject characteristics, and types of knee supports, direct comparisons with other studies should be made with caution. Limitations of our study include the lack of precise classification of injury types and severities. Therefore, it is not possible to draw conclusions about the effect of the brace on specific knee joint injuries, as the results reflect its impact on a broad spectrum of injury conditions. Future studies could focus on specific injury types to evaluate both the acute and long-term effects of different types of knee supports, including knee braces. Additionally, electromyographic analyses could provide deeper insights into neuromuscular control. This could also lead to better practical recommendations for the use of the brace, which is mainly intended for applications with higher loads, such as sports practice, especially under fatigued conditions.

## 5. Conclusions

In subjects with a history of knee joint injuries or pain, the injured leg showed various kinetic and kinematic differences compared to the healthy leg during running and jumping. Overall, the subjects appeared to reduce the magnitude and duration of knee joint loading on the injured leg, as indicated by lower maximal knee abduction moments, decreased vertical ground reaction forces, and shorter ground contact times during the stance phase of running. The injured leg also exhibited significantly reduced knee flexion during ground contact in jumping. 

Wearing the knee brace resulted in acute effects, such as more stabilized knee joint movement in the frontal plane. This was reflected by a significant reduction in maximal medio-lateral knee acceleration and a significant decrease in maximal knee abduction moment during running and jumping. The knee brace also normalized loading on the injured leg. Our results showed significantly higher maximal knee flexion moments, which were associated with higher vertical ground reaction forces, increased segment velocities, and greater maximal knee flexion angles. Subjects also reported less pain and felt more stable while wearing the knee brace. Therefore, we confirm our hypothesis that wearing a knee brace on the injured leg improves joint biomechanics by enhancing stability and kinematics while reducing pain during running and jumping, even in the presence of muscle fatigue. Consequently, wearing a knee brace after a knee joint injury may reduce the risk of reinjury.

## Figures and Tables

**Figure 1 bioengineering-12-00661-f001:**
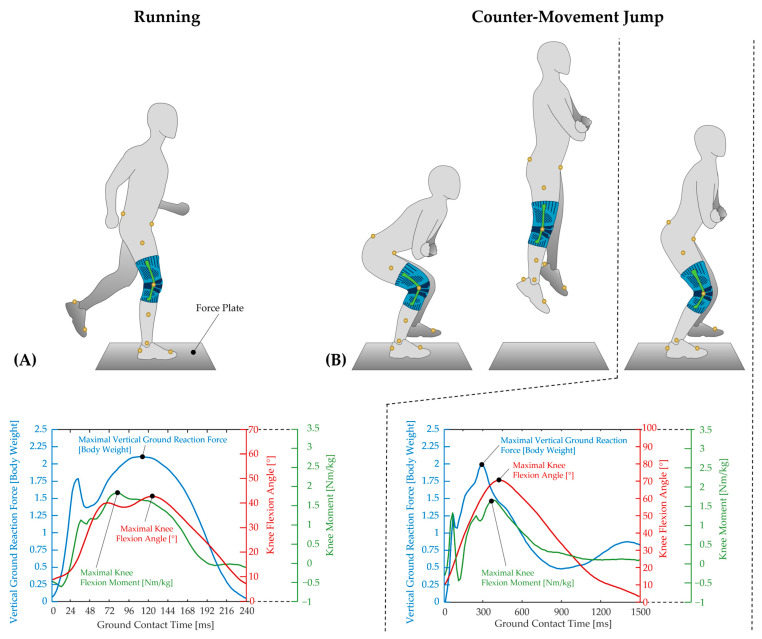
Subject performing the running (**A**) and jumping (**B**) tests on the force plate. For motion tracking 16 reflective markers (Plug-in Gait lower body marker set) were attached bilaterally to the pelvis (spina iliaca anterior superior and spina iliaca posterior superior), thighs, knee joints, tibiae, ankle joints, and toes. From the force plate and Vicon data, biomechanical parameters such as maximal vertical ground reaction force, ground contact time, maximal knee flexion angle, and maximal knee flexion moment were extracted. Note that for the knee moment, positive values represent flexion moments, while negative values indicate extension moments.

**Figure 2 bioengineering-12-00661-f002:**
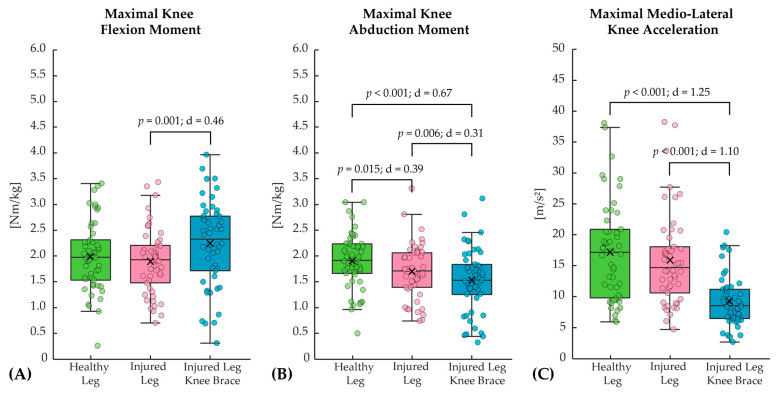
Comparison of the three measurement conditions (healthy leg, injured leg, and injured leg with knee brace) for the biomechanical parameters: (**A**) maximal knee flexion moment, (**B**) maximal knee abduction moment, and (**C**) maximal medio-lateral knee acceleration for the running test.

**Figure 3 bioengineering-12-00661-f003:**
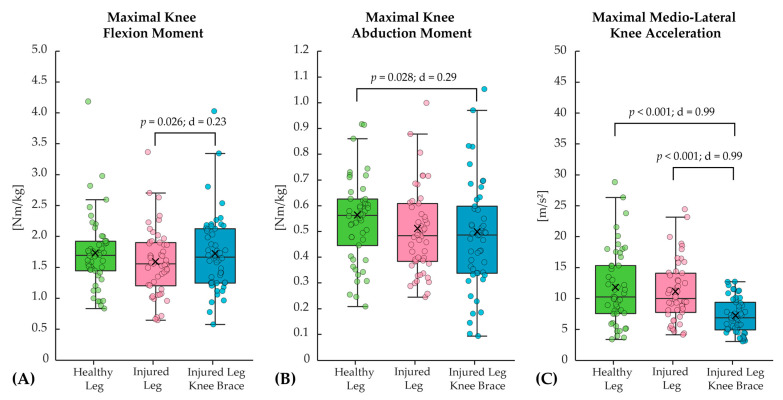
Comparison of the three measurement conditions (healthy leg, injured leg, and injured leg with knee brace) for the biomechanical parameters: (**A**) maximal knee flexion moment, (**B**) maximal knee abduction moment, and (**C**) maximal medio-lateral knee acceleration for the jumping test.

**Table 1 bioengineering-12-00661-t001:** Demographic and clinical data; presented as mean ± SD.

Age [years]	Height [cm]	Weight [kg]	Gender [Male/Female]	Physical Activity [Hours per Week]	Side of Injured Leg [Left/Right]	Time Since Injury [years]	Therapeutic Intervention [Conservative/Surgical]
33.5 ± 9.6	178.5 ± 9.7	74.4 ± 12.8	31/19	6.4 ± 4.0	28/22	6.1 ± 3.7	27/23
Types of Injuries of the Knee Joint							n (% of n)
Ligament Injuries: e.g., partial or complete tear of anterior and/or posterior cruciate ligaments, medial and/or lateral collateral ligaments							16 (32)
Meniscus Injuries: e.g., contusion or tear of the medial and/or lateral meniscus							16 (32)
Cartilage Damage: e.g., Patellofemoral and/or femoral-tibial cartilage damage or osteoarthritis (level 1)							13 (26)
Fractures and bone injuries: e.g., fractures of the patella or femur condyles							5 (10)
Inflammations and others: e.g., nonspecific load pain, tendinopathy, plica syndrome, edema							13 (26)

**Table 2 bioengineering-12-00661-t002:** Effects of the fatigue protocol on subjective rating and jumping height (mean ± SD). Statistically significant differences between conditions are indicated with a.

	Parameter	Before Fatigue Protocol	After Fatigue Protocol	*p*-Value	d
Fatigue Protocol	Fatigue (6–20)	7.2 ± 1.6 a	12.0 ± 2.7 a	a < 0.001	a = 0.73
	Maximal Jumping Height [cm]	35.2 ± 7.9 a	33.4 ± 7.9 a	a < 0.001	a = 0.23

**Table 3 bioengineering-12-00661-t003:** Effects of the knee brace on subjective ratings of pain and stability for the running and jumping tests (mean ± SD). Statistically significant differences between conditions are indicated with a.

	Parameter	Injured Leg	Injured Leg with Knee Brace	*p*-Value	d
Running	Pain (0–10)	2.1 ± 1.4 a	1.9 ± 1.3 a	a = 0.004	a = 0.15
	Stability (0–10)	2.0 ± 1.4	1.8 ± 1.3	0.125	-
Jumping	Pain (0–10)	2.2 ± 1.4	2.2 ± 1.4	1.000	-
	Stability (0–10)	2.0 ± 1.4 a	1.9 ± 1.3 a	0.070	-

**Table 4 bioengineering-12-00661-t004:** Comparison of the three measurement conditions (healthy leg, injured leg, and injured leg with knee brace) for the biomechanical parameters: maximal knee flexion angle, maximal vertical ground reaction force, and ground contact time for the running test (mean ± SD). Statistically significant differences between the three measurement conditions are indicated with a, b.

Parameter	Healthy Leg	Injured Leg	Injured Leg with Knee Brace	*p*-Value	d
Maximal Knee Flexion Angle [°]	43.7 ± 6.5	42.1 ± 6.4	43.6 ± 6.0	0.107	-
Maximal Vertical Ground Reaction Force [Body Weight]	2.43 ± 0.37	2.38 ± 0.34	2.43 ± 0.36	0.232	-
Ground Contact Time [ms]	258.2 ± 32.2 a	257.1 ± 33.1 b	253.8 ± 34.0 a; b	0.023	
				a = 0.038	a = 0.13
				b = 0.015	b = 0.10

**Table 5 bioengineering-12-00661-t005:** Comparison of the three measurement conditions (healthy leg, injured leg, and injured leg with knee brace) for the biomechanical parameters: maximal knee flexion angle, maximal vertical ground reaction force and jumping height for the jumping test (mean ± SD). Statistically significant differences between the three measurement conditions are indicated with a, b. Note that the parameter jumping height was only compared between the measurement conditions of the injured leg without and with brace.

Parameter	Healthy Leg	Injured Leg	Injured Leg with Knee Brace	*p*-Value	d
Maximal Knee Flexion Angle [°]	77.1 ± 23.2 a	74.4 ± 22.2 a; b	79.2 ± 22.9 b	a = 0.015 b < 0.001	a = 0.12 b = 0.21
Maximal Vertical Ground Reaction Force [Body Weight]	2.17 ± 0.87	2.04 ± 0.77	2.03 ± 0.85	0.162	-
Maximal Jumping Height [cm]	-	33.7 ± 6.8 a	33.2 ± 6.8 a	a = 0.007	a = 0.07

## Data Availability

The dataset used and analyzed in this study is available from the corresponding author upon reasonable request.

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
