# Peer review of "Running and Jumping After Muscle Fatigue in Subjects with a History of Knee Injury: What Are the Acute Effects of Wearing a Knee Brace on Biomechanics?"

_bioengineering, 2025, doi:10.3390/bioengineering12060661_

Round 1
Reviewer 1 Report
Comments and Suggestions for Authors
This study aims to analyze the acute effects of a knee bandage on lower limb kinematics and kinetics in individuals with a history of knee injury. The study appears very interesting and satisfactorily written. Congratulations to all the authors. However, it would benefit from some clarifications, improvements in terminology, and stronger integration with current literature.
- The abstract should be divided into the classic sections: background, methods, results and conclusion
- The conclusions, in the abstract as in the text, should be enriched with statistical information to give a clearer message.
- Clarify the mechanism of action hypothesized for this specific bandage (e.g., compression, proprioception, mechanical resistance) and define “biomechanics” operationally.
- Statistical test in Table 3 seems inappropriate: The pain and stability data (VAS) are ordinal, yet a t-test was applied.
- Phrases as “improved stability because of proprioception improvement” are causal claims that are unsupported without neurosensory assessments. I recommend using more cautious phrasing like “may reflect enhanced proprioception” or “might be due to”.
- Redundant References: Some references, as 16 and 20, are repeated multiple times in the same paragraph or sentence.
- “Bandage” may be misleading for English-speaking readers expecting something non-technical. Consider using “soft elastic knee sleeve” or define “bandage” clearly.
- Tables 3 and 4 include multiple p-values and letters without fully explaining annotation style (a, b, etc.).
- Some references are very up to date, others are not. Consider the role of the ACL in different contexts, especially in the introduction, citing for example these recent articles:
-- Mayer MA, Deliso M, Hong IS, et al. Rehabilitation and Return to Play Protocols After Anterior Cruciate Ligament Reconstruction in Soccer Players: A Systematic Review. Am J Sports Med. 2025 Jan;53(1):217-227. doi: 10.1177/03635465241233161.
--Gonarthrosis and ACL lesion: an intraoperative analysis and correlations in patients who underwent total knee arthroplasty. Passaretti A, Colò G, Bulgheroni A, et al. Minerva Orthopedics, 2024, 75(5), pp. 331–336. DOI 10.23736/S2784-8469.24.04425-0
- The rest of article seems to flow fluently and I congratulate the authors.
Author Response
Reviewer 1:
Open Review
(x) I would not like to sign my review report
( ) I would like to sign my review report Quality of English Language
( ) The English could be improved to more clearly express the research.
(x) The English is fine and does not require any improvement.
|
Yes |
Can be improved |
Must be improved |
Not applicable |
|
|
Does the introduction provide sufficient background and include all relevant references? |
( ) |
(x) |
( ) |
( ) |
|
Is the research design appropriate? |
(x) |
( ) |
( ) |
( ) |
|
Are the methods adequately described? |
(x) |
( ) |
( ) |
( ) |
|
Are the results clearly presented? |
(x) |
( ) |
( ) |
( ) |
|
Are the conclusions supported by the results? |
( ) |
(x) |
( ) |
( ) |
|
Are all figures and tables clear and well-presented? |
( ) |
(x) |
( ) |
( ) |
Comments and Suggestions for Authors
This study aims to analyze the acute effects of a knee bandage on lower limb kinematics and kinetics in individuals with a history of knee injury. The study appears very interesting and satisfactorily written. Congratulations to all the authors. However, it would benefit from some clarifications, improvements in terminology, and stronger integration with current literature.
- The abstract should be divided into the classic sections: background, methods, results and conclusion
Response: Thank you for the suggestion. However, according to the journal's author instructions, the abstract should not contain separate sections.
- The conclusions, in the abstract as in the text, should be enriched with statistical information to give a clearer message.
Response: Thank you for your comment. In the results and discussion sections, we already point out that the group differences were statistically significant. Therefore, we do not consider to repeat this in the conclusion. Nonetheless, we do mention in the conclusion of both the abstract and the main text that the differences were significant. Due to the word limit, it would also not be possible to repeat this in the abstract.
- Clarify the mechanism of action hypothesized for this specific bandage (e.g., compression, proprioception, mechanical resistance) and define “biomechanics” operationally.
Response: Thank you for your comment. We have added the specific mechanisms of action of the bandage in the introduction (line 48). Additionally, we have clarified the term "biomechanics" by specifying the related parameters.
- Statistical test in Table 3 seems inappropriate: The pain and stability data (VAS) are ordinal, yet a t-test was applied.
Response: Thank you for your valuable suggestion. You are right - for the two VAS scale parameters, pain and stability and the Borg scale parameter Fatigue we recalculated the statistics using a sign test, which is appropriate for ordinal data. The results have been updated accordingly in Table 2 and 3, and the sign test has been added to Section 2.3 Statistical Analysis (Line 147).
- Phrases as “improved stability because of proprioception improvement” are causal claims that are unsupported without neurosensory assessments. I recommend using more cautious phrasing like “may reflect enhanced proprioception” or “might be due to”.
Response: Thank you for your comment. We have qualified the statement regarding the relationship by using the word "might."
- Redundant References: Some references, as 16 and 20, are repeated multiple times in the same paragraph or sentence.
Response: Thank you for your comment. These sources were very helpful for our manuscript and are therefore cited multiple times. Nevertheless, we have removed one instance of duplicate citation.
- “Bandage” may be misleading for English-speaking readers expecting something non-technical. Consider using “soft elastic knee sleeve” or define “bandage” clearly.
Response: Thank you for your comment. We understand your point of view. The nature of the bandage is described in Section 2.2 Experimental Setup and Data Acquisition. Even the manufacturer uses the term "bandage" in English.
- Tables 3 and 4 include multiple p-values and letters without fully explaining annotation style (a, b, etc.).
Response: Thank you for the comment. We have added an explanation to the tables.
- Some references are very up to date, others are not. Consider the role of the ACL in different contexts, especially in the introduction, citing for example these recent articles:
Thank you for the comment. We have incorporated one of the sources you suggested at the appropriate place.
-- Mayer MA, Deliso M, Hong IS, et al. Rehabilitation and Return to Play Protocols After Anterior Cruciate Ligament Reconstruction in Soccer Players: A Systematic Review. Am J Sports Med. 2025 Jan;53(1):217-227. doi: 10.1177/03635465241233161.
--Gonarthrosis and ACL lesion: an intraoperative analysis and correlations in patients who underwent total knee arthroplasty. Passaretti A, Colò G, Bulgheroni A, et al. Minerva Orthopedics, 2024, 75(5), pp. 331–336. DOI 10.23736/S2784-8469.24.04425-0
- The rest of article seems to flow fluently and I congratulate the authors.
Thank you for your effort and time in reviewing the article.

Reviewer 2 Report
Comments and Suggestions for Authors
General characteristics and evaluation of the work:
The article addresses the clinically relevant topic of the effect of bandaging on the biomechanics of the knee joint in people with trauma and muscle fatigue, which is an important addition to the existing literature. The paper was correctly designed, using modern methods of measurement and adequate statistical analysis. The results are interesting and supported by both subjective and objective data. However, the article contains several important shortcomings that should be corrected before publication.
The paper is interesting and generally well-written but will require minor revisions and additions to both content and references before proceeding further. Below are my detailed notes and comments on the paper.
Minor comments:
The introduction is far too short and does not present the problem in sufficient detail.. Please expand the introduction with more detailed information on knee injuries, typical mechanism of injury and add the latest literature. I recommend adding the following literature items to the passage indicated: Comparison of diagnostic accuracy of physical examination and MRI in the most common knee injuries; Knee mri underestimates the grade of cartilage lesions;
The group's description lacks a detailed classification of injury types (e.g., cruciate ligament damage vs. meniscus), their severity and treatment method (conservative vs. surgical). Please elaborate on whether and how different types of injuries affected the analyzed parameters. It would be advisable to group the participants by type of injury and attempt a subgroup analysis.
The paper does not include EMG data, which could help understand the mechanisms of improved stabilization and changes in moments of force. The discussion should consider this lack as a limitation and propose recording EMG activity as part of future research.
The differences in pain and stability scores were statistically significant, but clinically very small (e.g., change in pain from 2.1 to 1.9). The authors should discuss the clinical significance of these differences and whether they have a real impact on patient function.
The findings suggest that banding improves biomechanics and reduces the risk of re-injury. Meanwhile, the study only looked at acute effects, with no long-term data. Phrases like “injury risk reduction” should be limited, and it should be noted that the results are only about immediate effects.
The paper is dominated by an analysis of the effects of one type of bandage, with no comparison with other solutions (rigid orthoses, dynamic orthoses, taping). Please expand the literature review in the introduction or discussion by referring to studies comparing different forms of knee joint support.
Measurement was performed on one limb only, under conditions of increasing fatigue. This may affect the variability of the data. The authors should discuss this aspect as a potential source of measurement error and limitation of the results.
I congratulate the authors on an interesting study and wish them further scientific success.
Author Response
Reviewer 2:
Open Review
(x) I would not like to sign my review report
( ) I would like to sign my review report Quality of English Language
( ) The English could be improved to more clearly express the research.
(x) The English is fine and does not require any improvement.
|
Yes |
Can be improved |
Must be improved |
Not applicable |
|
|
Does the introduction provide sufficient background and include all relevant references? |
( ) |
( ) |
(x) |
( ) |
|
Is the research design appropriate? |
( ) |
(x) |
( ) |
( ) |
|
Are the methods adequately described? |
( ) |
(x) |
( ) |
( ) |
|
Are the results clearly presented? |
( ) |
( ) |
(x) |
( ) |
|
Are the conclusions supported by the results? |
( ) |
(x) |
( ) |
( ) |
|
Are all figures and tables clear and well-presented? |
( ) |
(x) |
( ) |
( ) |
Comments and Suggestions for Authors
General characteristics and evaluation of the work:
The article addresses the clinically relevant topic of the effect of bandaging on the biomechanics of the knee joint in people with trauma and muscle fatigue, which is an important addition to the existing literature. The paper was correctly designed, using modern methods of measurement and adequate statistical analysis. The results are interesting and supported by both subjective and objective data. However, the article contains several important shortcomings that should be corrected before publication.
The paper is interesting and generally well-written but will require minor revisions and additions to both content and references before proceeding further. Below are my detailed notes and comments on the paper.
Minor comments:
The introduction is far too short and does not present the problem in sufficient detail.. Please expand the introduction with more detailed information on knee injuries, typical mechanism of injury and add the latest literature. I recommend adding the following literature items to the passage indicated: Comparison of diagnostic accuracy of physical examination and MRI in the most common knee injuries; Knee mri underestimates the grade of cartilage lesions;
Response: Thank you for your comment. Due to the large number of potentially affected structures, including ligaments, menisci, cartilage, and bones, we intentionally did not describe the diverse injury mechanisms in detail, in order to maintain the focus of the introduction. Explanations regarding these mechanisms are instead provided in the discussion section. The main focus of our study was not on the causes or mechanisms of multiple knee injuries, but rather on the effects of the bandage on biomechanics, pain and joint instability. Regarding the suggested paper, which investigates the diagnostic accuracy of physical examination and MRI in knee injuries, we believe it does not directly align with the content of the manuscript.
The group's description lacks a detailed classification of injury types (e.g., cruciate ligament damage vs. meniscus), their severity and treatment method (conservative vs. surgical). Please elaborate on whether and how different types of injuries affected the analyzed parameters. It would be advisable to group the participants by type of injury and attempt a subgroup analysis.
Response: Thank you for your comment. Unfortunately, we did not receive more detailed information about the participants' injuries. Since many participants had injuries involving multiple structures, a more specific classification would have resulted in a smaller sample sized subgroups and reduced statistical power. Since knee bandages are generally used for a wide range of knee-related issues, including non-specific knee pain and joint instability, our aim was to investigate a relatively heterogeneous group. However, we understand that a more precise classification would allow for more specific statements about the parameters related to particular injuries and could lead to more targeted recommendations. We will take this valuable suggestion into account for future studies and aim to recruit numerous participants with more uniform injury profiles.
The paper does not include EMG data, which could help understand the mechanisms of improved stabilization and changes in moments of force. The discussion should consider this lack as a limitation and propose recording EMG activity as part of future research.
The statement is included in the manuscript (line 322): Additionally, electromyographic analyses could provide deeper insights into neuromuscular control.
The differences in pain and stability scores were statistically significant, but clinically very small (e.g., change in pain from 2.1 to 1.9). The authors should discuss the clinical significance of these differences and whether they have a real impact on patient function.
Thank you for your comment. We have added a sentence (line 304) that puts the relevance of the biomechanical and subjective parameters into context.
The findings suggest that banding improves biomechanics and reduces the risk of re-injury. Meanwhile, the study only looked at acute effects, with no long-term data. Phrases like “injury risk reduction” should be limited, and it should be noted that the results are only about immediate effects.
Response: Thank you for your comment. We added “acute effects” in the conclusion section (line 339).
The paper is dominated by an analysis of the effects of one type of bandage, with no comparison with other solutions (rigid orthoses, dynamic orthoses, taping). Please expand the literature review in the introduction or discussion by referring to studies comparing different forms of knee joint support.
Response: Thank you for your comment. As we already described in the limitations section, a direct comparison with other study results using different knee supports, etc., is, in our opinion, not possible due to differences in methodology and study populations. Our aim was also to investigate the acute effect of this particular bandage, rather than to compare it with other types of knee supports.
Measurement was performed on one limb only, under conditions of increasing fatigue. This may affect the variability of the data. The authors should discuss this aspect as a potential source of measurement error and limitation of the results.
Our subjects had unilateral injuries, and the bandage was worn only on the injured leg. Both legs (injured and healthy) were involved in the measurements and the fatigue protocol. However, all trials were randomized to minimize potential effects of fatigue or recovery.
I congratulate the authors on an interesting study and wish them further scientific success.
Thank you for your effort and time in reviewing the article.

Reviewer 3 Report
Comments and Suggestions for Authors
The article addresses the important topic of the biomechanical effects of knee bracing in people with a history of knee injury. The authors conducted a study involving 50 subjects who had undergone unilateral knee injury. They analyzed biomechanical parameters during running and jumping, under conditions of muscle fatigue, comparing the healthy limb, the injured limb and the injured limb with the applied brace. The work is interesting and with minor corrections can be further processed.
Minor comments:
1. The authors should supplement demographic and clinical data with a detailed breakdown of injury types (e.g., cartilage damage, meniscus ruptures, enthesopathies) and determine their severity and current functional status (e.g., based on the IKDC or Tegner scale). I also recommend supplementing the information on the most common types of injuries to structures above with the addition of references. I recommend adding the following literature items to the passage indicated: DOI 10.1088/1742-6596/1736/1/012027 ; DOI 10.3390/healthcare12161648
2. The authors should expand the discussion section to include practical clinical recommendations for the use of the brace, taking into account different patient groups, phases of rehabilitation, and potential risks associated with long-term use.
3. In order to more fully put the results in the context of existing knowledge, it is worth comparing the applied brace with other types of knee joint support, especially in the context of dynamic activities. This will enhance the value of the work.
4. In the discussion, it is worth noting that the low level of pain (average 2/10) may limit the ability to demonstrate the full effects of the intervention and may not reflect the situation of patients in the more acute post-traumatic stage.
Consideration of the above recommendations will significantly increase the scientific and practical value of the work.
Author Response
Reviewer 3:
Open Review
( ) I would not like to sign my review report
(x) I would like to sign my review report Quality of English Language
( ) The English could be improved to more clearly express the research.
(x) The English is fine and does not require any improvement.
|
Yes |
Can be improved |
Must be improved |
Not applicable |
|
|
Does the introduction provide sufficient background and include all relevant references? |
( ) |
(x) |
( ) |
( ) |
|
Is the research design appropriate? |
( ) |
(x) |
( ) |
( ) |
|
Are the methods adequately described? |
( ) |
(x) |
( ) |
( ) |
|
Are the results clearly presented? |
( ) |
(x) |
( ) |
( ) |
|
Are the conclusions supported by the results? |
(x) |
( ) |
( ) |
( ) |
|
Are all figures and tables clear and well-presented? |
(x) |
( ) |
( ) |
( ) |
Comments and Suggestions for Authors
The article addresses the important topic of the biomechanical effects of knee bracing in people with a history of knee injury. The authors conducted a study involving 50 subjects who had undergone unilateral knee injury. They analyzed biomechanical parameters during running and jumping, under conditions of muscle fatigue, comparing the healthy limb, the injured limb and the injured limb with the applied brace. The work is interesting and with minor corrections can be further processed.
Minor comments:
1. The authors should supplement demographic and clinical data with a detailed breakdown of injury types (e.g., cartilage damage, meniscus ruptures, enthesopathies) and determine their severity and current functional status (e.g., based on the IKDC or Tegner scale). I also recommend supplementing the information on the most common types of injuries to structures above with the addition of references. I recommend adding the following literature items to the passage indicated: DOI 10.1088/1742-6596/1736/1/012027 ; DOI 10.3390/healthcare12161648
Response: Thank you for your comment. The information about the injuries, as well as most types of injuries, is presented in Table 1. Unfortunately, we do not have any additional information, and we plan to include the IKDC in our next study. Regarding the suggested papers, we believe they do not directly align with the content of the manuscript.
- The authors should expand the discussion section to include practical clinical recommendations for the use of the brace, taking into account different patient groups, phases of rehabilitation, and potential risks associated with long-term use.
Response: Thank you for your comment. Based on our results, a recommendation can be made, which we have added in line 323.
- In order to more fully put the results in the context of existing knowledge, it is worth comparing the applied brace with other types of knee joint support, especially in the context of dynamic activities. This will enhance the value of the work.
Response: Thank you for your comment. As we already described in the limitations section, a direct comparison with other study results using different knee supports, etc., is, in our opinion, not possible due to differences in methodology and study populations. To our knowledge, no study has investigated these effects under conditions of muscular fatigue, which further complicates comparisons. Our aim was also to investigate the acute effect of this particular bandage, rather than to compare it with other types of knee supports. Nevertheless, we appreciate these suggestions for future research ideas.
- In the discussion, it is worth noting that the low level of pain (average 2/10) may limit the ability to demonstrate the full effects of the intervention and may not reflect the situation of patients in the more acute post-traumatic stage.
Consideration of the above recommendations will significantly increase the scientific and practical value of the work.
Response: Thank you for your comment. The requirement for participation in the study was a verbal or written return-to-sport recommendation from a medical doctor and the ability of the participants to complete the fatigue protocol. This was communicated during a preliminary phone call with the subjects. As a result, only participants with relatively low pain levels due to injuries sustained some time ago (Table 1: time since injury: 6.1 ± 3.7 years) volunteered and were included in the study. Nevertheless, thank you very much for the valuable suggestions, which could be considered in future studies.
Thank you for your effort and time in reviewing the article.
